# Machine Learning-Based Image Reconstruction in Wearable CC-EIT of the Thorax: Robustness to Electrode Displacement

**DOI:** 10.3390/s25216543

**Published:** 2025-10-23

**Authors:** Jan Jeschke, Mikhail Ivanenko, Waldemar T. Smolik, Damian Wanta, Mateusz Midura, Przemysław Wróblewski

**Affiliations:** Faculty of Electronics and Information Technology, Warsaw University of Technology, Nowowiejska 15/19, 00-665 Warsaw, Poland; jan.jeschke.stud@pw.edu.pl (J.J.); mikhail.ivanenko.dokt@pw.edu.pl (M.I.); waldemar.smolik@pw.edu.pl (W.T.S.); mateusz.midura@pw.edu.pl (M.M.); przemyslaw.wroblewski@pw.edu.pl (P.W.)

**Keywords:** capacitively coupled electrical impedance tomography, image reconstruction, inverse problem, deep neural networks, deep learning, fully connected neural networks, cGAN, medical imaging, lung imaging, pneumothorax, pleural effusion

## Abstract

This study investigates the influence of variable electrode positions on image reconstruction in capacitively coupled electrical impedance tomography (CC-EIT) of the human thorax. Images were reconstructed by an adversarial neural network trained on a synthetic dataset generated using a tomographic model that included a wearable elastic band with 32 electrodes attached. Dataset generation was conducted using a previously developed numerical phantom of the thorax, combined with a newly developed algorithm for random selection of electrode positions based on physical limitations resulting from the elasticity of the band and possible position inaccuracies while putting the band on the patient’s chest. The thorax phantom included the heart, lungs, aorta, and spine. Four training and four testing datasets were generated using four different levels of electrode displacement. Reconstruction was conducted using four versions of neural networks trained on the datasets, with random ellipses included and noise added to achieve an SNR of 30 dB. The quality was assessed using pixel-to-pixel metrics such as the root-mean-square error, structural similarity index, 2D correlation coefficient, and peak signal-to-noise ratio. The results showed a strong negative influence of electrode displacement on reconstruction quality when no samples with displaced electrodes were present in the training dataset. Training the network on the dataset containing samples with electrode displacement allowed us to significantly improve the quality of the reconstructed images. Introducing samples with misplaced electrodes increased neural network robustness to electrode displacement while testing.

## 1. Introduction

Electrical capacitance tomography (ECT) is a non-invasive method for discovering the inner structure of an object with spatially variable electrical properties [1,2]. It aims to recover electrical permittivity and conductivity distribution inside a field of view using capacitance measurements made by electrodes surrounding an object. ECT is broadly used in production for monitoring dynamic processes such as combustion [3,4] or multi-phase flow [5,6]. Despite its low spatial resolution, ECT has very high temporal resolution, which makes it very useful in the surveillance of systems prone to rapid changes. Usually, a tomographic sensor is built as a rigid ring of electrodes surrounded by a screen, where the inner part of the ring becomes the field of view [7,8]. This device construction is very useful for monitoring different kinds of flow in pipes, because constant geometry simplifies the procedure of reconstructing the distribution of electrical properties.

However, in recent years, interest has arisen in the medical usage of ECT due to its non-invasiveness. Electrical tomography does not rely on any potentially harmful radiation, which makes it very promising for analyzing the inner structure of living beings. For example, due to the difference in the high electrical conductivity of lung tissue between the inhale and exhale phases, it allows researchers to conduct lung ventilation monitoring [9]. Moreover, it is possible to detect diseases characterized by abrupt changes in electrical conductivity, such as pneumothorax or pleural effusion [10]. Today, there is a variety of medical devices (e.g., Dräger PulmoVista [11]) that rely on electrical impedance tomography, which is based on conducting impedance measurements. This measuring method requires maintaining as low a skin–electrode impedance as possible, which is usually achieved by gluing electrodes to the patient’s skin. Measuring capacitance, on the other hand, makes it possible to use so-called dry electrodes [12], because capacitance is not strongly influenced by skin–electrode impedance. Traditional ECT is aimed at recovering electrical permittivity distribution in the field of view; however, its medical application is usually focused on finding the conductivity distribution. Therefore, it is referred to as capacitively coupled electrical impedance tomography (CC-EIT) [13,14,15,16].

Producing an electrical property distribution map based on capacitance measurements is called inverse problem solving [17] or image reconstruction. Throughout the years of research on electrical tomography, a broad variety of inverse problem solving methods have been developed [18]. The traditional approach is the algebraic one, where the forward tomographic process is represented by a vector function that takes the electrical property distribution as an input and the capacitance measurement vector as an output. The inverse problem solution in that case is supposed to be the inverse of that function. From a mathematical point of view, this problem is severely ill-posed because the size of the vector representing the electrical property distribution is much larger than the number of capacitance measurements [19].

Despite the process non-linearity, the first class of reconstruction methods relies on the linear approximation of the problem. Among such methods are linear back projection (LBP), which is a one-step method, and the iterative Landweber algorithm [20]. They are pretty fast, specifically LBP, but the reconstruction results are quite poor. The best possible reconstruction results can be obtained using non-linear iterative methods such as the nonlinear Landweber [21] or Levenberg–Marquardt (LM) algorithm [22]. However, despite satisfactory reconstruction quality, non-linear methods are very slow due to their computational complexity. There are some simplifications of the LM algorithm, which allow for reducing the reconstruction time [23]. However, the main challenge is the necessity to numerically solve the differential electric field equation at one of the steps, which employs a matrix inverse operation, and it is very difficult to speed up.

Algebraic reconstruction methods’ challenges caused a constant search for other ways to solve an inverse problem in electrical tomography. Rapid progress in machine learning (ML) gave the idea to try its methods for image reconstruction [24,25,26]. The most promising approach was the usage of artificial neural networks (ANNs) to reconstruct images from the capacitance measurements [27]. ANNs are non-linear by definition, and this makes them suitable to represent the tomographic process. The simplest way to train the neural network is supervised training, which uses samples of electrical conductivity distribution and corresponding capacitance measurements. Providing a sufficiently large training dataset, it is possible to achieve satisfactory reconstruction quality. Also, due to negligible inference time, such a network can be used in near real time to obtain electrical properties’ distribution maps from constantly conducted capacitance measurements. However, there are still some challenges to overcome. Among them are network architecture selection and obtaining a suitable training dataset. Our previous research showed auspicious results when using UNet-based ANNs [28] trained by employing a conditional generative adversarial (cGAN) approach [10]. However, satisfactory results can also be achieved by using fully connected networks [29].

In contrast to industrial applications, where a tomographic sensor can be rigid, in medical applications, the sensor should be made as an elastic band with electrodes attached. This band can be worn on the patient’s body, making the electrodes stick to the body. In the previous study, we described an approach to model electrical tomography of the human thorax, assuming a fixed geometry [10]. However, in real life, a band’s elasticity implies a changing geometry, and image reconstruction methods in ECT assume known sensor geometry. Therefore, it is expected that the movement of electrodes during data acquisition should hinder the reconstruction.

The issue of moving electrodes was addressed in the context of electrical impedance tomography in a number of studies. One of the earliest works [30] explored the influence of sensor size changes on image reconstruction. However, during the simulation, only the uniform translation was considered. In the other remarkable study [31], the authors verified how the electrodes’ positioning inaccuracy affects EIT image reconstruction during real measurements in humans. They found that breathing systematically influences EIT results. However, the number of subjects was too small to make general conclusions. Considering electrode displacement while using machine learning methods was investigated in [32]. However, the authors only managed to detect the fact of electrode positioning error, but did not take it into account during image reconstruction. To the best of our knowledge, there was no attempt to analyze electrode misplacement in the context of electrical capacitance tomography. Additionally, there was no attempt to use neural networks for image reconstruction in ECT with displaced electrodes.

Therefore, the motivation of this paper was to explore the influence of electrode displacement on image reconstruction quality. To achieve that, it was necessary to prepare a training dataset, including the samples with disturbances in the tomographic sensor position, assuming the electrodes are attached to the elastic band.

The main challenge was to develop a training dataset generation algorithm, taking into account the following variable factors:position and size of the organs;gaps between the electrodes;sensor rotation relative to the patient’s body;noise in the measurement data.

To achieve that, it was possible to reuse a previously created model of the standard thorax, which assumes fixed electrode positions. In addition, it was necessary to develop a mathematical model of electrode position changes caused by the elasticity of the band and by sensor rotation resulting from inaccuracies in attaching a band to the patient’s body. The existing model was developed using ECTSim 2024 v 1.0 toolbox for Matlab 2024 [33]. This model is aimed at simulating electrical capacitance tomography of the thorax using 32 electrodes. It includes only the organs that have a significant difference in electrical conductivity, such as lungs, heart, aorta, and spine. Additionally, it is possible to simulate two diseases affecting the conductivity of the lungs, such as pneumothorax, which lowers the conductivity due to excessive air in the pleural cavity, and pleural effusion, which is characterized by increased conductivity due to the abnormal accumulation of fluid in the pleural space [10].

The numerical model of the human thorax represents the internal organs mentioned above as a combination of elliptic regions. Such a numerical phantom is suitable for the generation of a large number of samples with variable organ positions and shapes. It is designed to produce 64 × 64 maps of the electrical conductivity using 992 measurements from 32 electrodes.

Adopting variable geometry requires the development of an algorithm capable of randomly assigning electrode positions according to the defined limits resulting from the physics of the elastic band, where electrodes are attached, and overall band position inaccuracies while attaching it to the patient’s body. When putting a band on the patient’s chest, it is impossible to achieve complete precision of the angular band’s position. Such inaccuracy will result in the band’s rotation. The elasticity of the band will cause the electrodes to shift relative to each other.

It is supposed that introducing samples with displaced electrodes into the training dataset should lead to an increase in the network’s robustness to electrode displacement at the inference phase. It should allow for mitigating the negative effects of electrode shifting and obtaining images with satisfactory quality.

## 2. Materials and Methods

This work is mainly focused on exploring the influence of electrode shifting in CC-EIT on the quality of human thorax image reconstruction, conducted by an artificial neural network. Training a neural network requires obtaining a large dataset of CC-EIT measurements and corresponding electrical conductivity distributions. Building a dataset using real measurements is a substantial engineering challenge, but we can simulate measurements using a numerical phantom of the human thorax from our previous study. In this paper, we continue our research by developing an algorithm capable of generating physically possible movements of electrodes attached to the elastic band placed on the patient’s chest.

### 2.1. Numerical Simulation of CC-EIT Measurements

A tomographic sensor in CC-EIT usually consists of a number of electrodes surrounding the examined object, and it is covered by a screen. During the measurement, the voltage is applied to one of the electrodes while maintaining zero potential on the other electrodes and a screen. In CC-EIT, the distribution of the electrical properties can be represented by a complex electrical permittivity: (1)$$\varepsilon =\varepsilon^{\prime }+j\frac{\sigma }{\omega },$$
where $\varepsilon^{\prime }$ is permittivity, σ is conductivity, and ω is angular frequency. Using a known potential as a boundary condition, it is possible to calculate the potential distribution by solving an electric field differential equation: (2)∇·ε∇φ=0,
where ε is permittivity, and φ is potential. Applying the finite volume method [34] allows for solving that equation numerically. It is necessary to calculate a potential distribution for all cases when each of the electrodes has an applied voltage. After obtaining all values of the potential, it is possible to find elements of the so-called sensitivity matrix, which shows how changes in permittivity at a given point of the grid influence changes in each capacitance measurement by employing the reciprocity principle [35]: (3)$$\frac{\partial C_{AB}}{\partial \varepsilon_{n}}=-\frac{1}{U^{2}}\int_{\Omega_{n}}\nabla \varphi_{A}\nabla \varphi_{B}d\Omega_{n},$$
where $C_{AB}$—capacitance between electrodes A and B, $\varepsilon_{n}$—electrical permittivity in the n-th small volume element, U—voltage applied between electrodes, $\Omega_{n}$ is a small area space element, $\varphi_{A}$, $\varphi_{B}$—electrical potentials when the electrode A or B is an excitation electrode, respectively.

The calculated sensitivity matrix can be further used to obtain capacitance measurements from a given complex electrical permittivity distribution by applying a linear approximation: (4)$$C_{M}=S_{M\times N}\varepsilon_{N},$$
where *S* is sensitivity matrix, having elements obtained by (Equation 3), *C*—capacitance measurements, ε—complex electrical permittivity, *M* is number of mutual capacitance measurements, and *N* is total number of pixels where electrical permittivity is given.

To conduct calculations, we used the ECTSim toolkit for Matlab 2024 [33]. It allows defining the regions having different electrical properties with the help of set arithmetic, making it possible to combine simple geometric shapes like ellipses and rectangles. The toolkit has convenient functions for each of the measurement calculation steps described above. It uses the finite volume method to solve the Gauss law differential Equation (Equation 2) and the reciprocity principle (Equation 3) to calculate elements of the sensitivity matrix. Measurement simulation conducted in such a way ensures that subsequent neural network training is performed on data that implicitly includes electromagnetic priors reflecting electrical field behavior inside the tomographic sensor.

### 2.2. Numerical Phantom of the Human Thorax

In the case of human thorax imaging, the sensor can be made in the form of an elastic band attached to the patient’s chest, as shown in Figure 1. A design for the sensor was proposed and discussed in many works within the context of electrical impedance tomography utilizing active electrodes [36,37,38,39], as well in the context of CC-EIT using active dry electrodes [40,41]. The common approach shown in the mentioned works is to use active electrodes. Another possibility is to add a screen, ensuring the electrodes are sensitive only to the electric field inside the sensor.

In CC-EIT, it is necessary to use advanced capacitance measurement techniques to ensure high precision. In our lab we use the EVT4 data acquisition system, which employs a single shot high-voltage measurement method. It allows achieving a signal-to-noise ratio of 30 dB for the capacitance value level corresponding to the opposite electrodes. Generally, it is impossible to eliminate all noise sources. However, our previous research showed that it is justifiable to use a Gaussian noise approach while modeling CC-EIT data acquisition [42].

A model of a 2D EIT slice of the human thorax was first proposed in [43]. It was based on the axial lung computed tomography (CT) slice segmentation [44]. The CT image was taken when the patient was in the supine position. The phantom includes the most distinctive organs, such as the heart, aorta, spine, and lungs. The enhanced model, proposed in [10], allows for simulation of two health conditions affecting lung conductivity, alongside healthy lungs: pneumothorax and pleural effusion (Figure 2).

### 2.3. Generation of Inaccuracies in the Electrode Position

The numerical phantom of the human thorax, described above, has fixed electrode positions. However, in real life, attaching an electrode belt to a patient’s chest inevitably leads to some inaccuracies in the belt’s position relative to the patient’s body. Therefore, it is crucial to find out what the theoretical limits of electrode shifting are that still allow for obtaining satisfactory image reconstruction results. There are two main sources of electrode position variations. The first source is an inaccuracy in the first electrode position relative to the patient’s body. Assuming the ring of the sensor is rigid, this first electrode misposition will lead to the sensor’s rotational shift, which can be expressed in degrees. However, belt flexibility introduces the other component of the shift. When the sensor’s rotation keeps the distance between electrodes, the belt’s stretching shifts the electrodes relative to each other.

The described human thorax phantom introduces a so-called standard patient. It has a fixed thorax perimeter that simplifies a numerical simulation. However, in the real world, the size of a thorax varies between patients, and even within the same patient, it varies due to the respiratory activity. This implies that using a static model to reconstruct images will lead to image distortion, specifically changes in the internal organ sizes and shapes. For example, imaging a patient with a larger thorax size than that adopted in the model should cause a reduction in the size of the heart, aorta, and spine on the image. Such changes are taken into account by varying those sizes and shapes during numerical simulation.

To simulate the belt’s flexibility, it is possible to employ a simple elastic deformation model [45] governed by laws of physics. However, fixing the belt length in the adopted model leads to apparent non-physical belt deformation and requires another modeling approach. Such an approach should be capable of generating broader electrode displacements than allowed by the real-world elastic deformation laws. It is possible to use a function that maps the distance from the first electrode in the rigid case to the distance in the stretched case. We can assume that the position of the first electrode is zero and the position of the last electrode is one. It is true for both rigid and stretched cases. Additionally, the electrodes cannot overlap, which means the position of the next electrode should always be greater than the position of the current electrode. Therefore, the mapping function should map zero to zero and one to one, and it should always be increasing at the interval from zero to one: (5)f:[0;1]→[0;1],(6)f:↗[0;1],

The trivial case of such a function is $f\left(x\right)=x$, which corresponds to the rigid case. One of the families of functions, which can meet the given condition, is the sum of the sine functions rotated by a $\frac{\pi }{4}$ around the center of the coordinate system. Such a sum can be given by a parametric equation: (7)$$\left(x\left(t\right),y\left(t\right)\right)=\left(t,\sum_{i=1,2\ldots }a_{i}sin\left(i\pi t+b_{i}\right)\right),$$

Using a rotation matrix, we can describe a rotated curve by: (8)$$\left[\begin{array}{c} x_{1}\left(t\right)\\ y_{1}\left(t\right) \end{array}\right]=\left[\begin{array}{cc} cos\frac{\pi }{4} & -sin\frac{\pi }{4}\\ sin\frac{\pi }{4} & cos\frac{\pi }{4} \end{array}\right]\left[\begin{array}{c} x(t)\\ y(t) \end{array}\right]=\left[\begin{array}{c} \frac{\sqrt{2}}{2}\left(t-\sum_{i=1,2\ldots }a_{i}sin\left(i\pi t+b_{i}\right)\right)\\ \frac{\sqrt{2}}{2}\left(t+\sum_{i=1,2\ldots }a_{i}sin\left(i\pi t+b_{i}\right)\right) \end{array}\right],$$

To simplify calculations, we can define: (9)$$\left[\begin{array}{c} x_{2}\left(t\right)\\ y_{2}\left(t\right) \end{array}\right]=\sqrt{2}\left[\begin{array}{c} x_{1}\left(t\right)\\ y_{1}\left(t\right) \end{array}\right]=\left[\begin{array}{c} t-\sum_{i=1,2\ldots }a_{i}sin\left(i\pi t+b_{i}\right)\\ t+\sum_{i=1,2\ldots }a_{i}sin\left(i\pi t+b_{i}\right) \end{array}\right],$$

From condition (Equation 5), we can state that $t_{i}$ and $t_{j}$ exist as follows: (10)$$\exists t_{i}:x_{2}\left(t_{i}\right)=y_{2}\left(t_{i}\right)=0;\exists t_{j}:x_{2}\left(t_{j}\right)=y_{2}\left(t_{j}\right)=1,$$

We can notice that for any function g(t), it is possible to find a first-degree polynomial $h_{g}\left(t\right)=a_{h}t+b_{h}$, which meets the following conditions: (11)$$g\left(0\right)+h_{g}\left(0\right)=0;g\left(1\right)+h_{g}\left(1\right)=1,$$

Assuming $\hat{g}\left(t\right)=g\left(t\right)+h_{g}\left(t\right)$ we can meet the condition (Equation 10) by substituting $x_{2}\left(t\right)$ by ${\hat{x}}_{2}\left(t\right)$ and $y_{2}\left(t\right)$ by ${\hat{y}}_{2}\left(t\right)$. It is possible to find $a_{h}$ and $b_{h}$ by solving the following system of equations: (12)$$\left\{\begin{array}{c} \hat{g}\left(0\right)=0\\ \hat{g}\left(1\right)=1 \end{array}\right.,$$

Using the definition of $\hat{g}\left(t\right)$ and $h_{g}\left(t\right)$, the system can be written as follows: (13)$$\left\{\begin{array}{c} 0=\hat{g}\left(0\right)=g\left(0\right)+h_{g}\left(0\right)=g\left(0\right)+a_{h}\;\cdot \;0+b_{h}=g\left(0\right)+b_{h}\\ 1=\hat{g}\left(1\right)=g\left(1\right)+h_{g}\left(1\right)=g\left(1\right)+a_{h}\;\cdot \;1+b_{h}=g\left(0\right)+{a_{h}+b}_{h} \end{array}\right.$$

Therefore: (14)$$\left\{\begin{array}{c} b_{h}=-g\left(0\right)\\ a_{h}=1-g\left(1\right)-b_{h} \end{array}\right.$$(15)$$\left\{\begin{array}{c} b_{h}=-g\left(0\right)\\ a_{h}=1-g\left(1\right)+g\left(0\right) \end{array}\right.$$

Applying the solution (Equation 15) to the definition of $h_{g}\left(t\right)=a_{h}t+b_{h}$, we get: (16)$$h_{g}\left(t\right)=a_{h}t+b_{h}=\left(1-g\left(1\right)+g\left(0\right)\right)\;\cdot \;t-g\left(0\right)=t\left(1-g\left(1\right)\right)+\left(t-1\right)g\left(0\right)$$
and then we can define the mapping from g(t) to $\hat{g}\left(t\right)$ as: (17)$$g\left(t\right)\to \hat{g}\left(t\right)=g\left(t\right)+t\left(1-g\left(1\right)\right)+\left(t-1\right)g\left(0\right)$$

By applying the mapping (Equation 17) to the Equation (Equation 9), we can define: (18)$$\left[\begin{array}{c} x_{3}\left(t\right)\\ y_{3}\left(t\right) \end{array}\right]=\left[\begin{array}{c} x_{2}\left(t\right)+\left(t-1\right)\;\cdot \;x_{2}\left(0\right)+t\;\cdot \;\left(1-x_{2}\left(1\right)\right)\\ y_{2}\left(t\right)+\left(t-1\right)\;\cdot \;y_{2}\left(0\right)+t\;\cdot \;\left(1-y_{2}\left(1\right)\right) \end{array}\right]=\left[\begin{array}{c} t-S(t)\\ t+S(t) \end{array}\right]$$
where function *S* is defined as: (19)$$\begin{array}{c} S\left(t\right)=\sum_{i=1,3,5,\ldots }a_{i}\left(sin\left(i\pi t+b_{i}\right)+\left(2t-1\right)sin\left(b_{i}\right)\right)\\ +\sum_{i=2,4,6,\ldots }a_{i}\left(sin\left(i\pi t+b_{i}\right)-sin\left(b_{i}\right)\right) \end{array}$$

The always increasing condition on the function *f* is equal to the condition of the first derivative to be positive on the whole interval from zero to one. We can calculate the first derivative as: (20)$$f^{\prime }\left(x\right)=\frac{dy_{3}\left(t\right)}{dx_{3}\left(t\right)}=\frac{\frac{d}{dt}\left(t+S\left(t\right)\right)}{\frac{d}{dt}\left(t-S\left(t\right)\right)}=\frac{1+S^{\prime }\left(t\right)}{1-S^{\prime }\left(t\right)}$$
and obtain inequality: (21)$$\frac{1+S^{\prime }\left(t\right)}{1-S^{\prime }\left(t\right)}>0\Rightarrow \left(1+S^{\prime }\left(t\right)\right)\left(1-S^{\prime }\left(t\right)\right)>0\Rightarrow 1-{\left(S^{\prime }\left(t\right)\right)}^{2}>0\Rightarrow {\left(S^{\prime }\left(t\right)\right)}^{2}<1$$

When meeting this inequality, both conditions (Equation 5) and (Equation 6) will be met as well. Finding parameters for all i∈N would give an infinite number of parameters. Therefore, it is necessary to limit the value of *i* in some way. In this work, we decided to use only i=1,2, and that will give us a family of functions defined by only four parameters: (22)$$\left[\begin{array}{c} x_{3}\left(t\right)\\ y_{3}\left(t\right) \end{array}\right]=\left[\begin{array}{c} t-a_{1}\left(sin\left(\pi t+b_{1}\right)+\left(2t-1\right)sin\left(b_{1}\right)\right)-a_{2}\left(sin\left(2\pi t+b_{2}\right)-sin\left(b_{2}\right)\right)\\ t+a_{1}\left(sin\left(\pi t+b_{1}\right)+\left(2t-1\right)sin\left(b_{1}\right)\right)+a_{2}\left(sin\left(2\pi t+b_{2}\right)-sin\left(b_{2}\right)\right) \end{array}\right]$$
which would be limited by the following inequality: (23)$$|S_{2}^{\prime }\left(t\right)|=|a_{1}\pi cos\left(\pi t+b_{1}\right)+a_{2}2\pi cos\left(2\pi t+b_{2}\right)+2a_{1}sin\left(b_{1}\right)|\leq 1$$

It is possible to calculate the maximum value of $|S_{2}^{\prime }\left(t\right)|$ using trigonometric function limits cos(x)≤1 and sin(x)≤1, and triangle inequality |x+y|≤|x|+|y|: (24)$$\begin{array}{c} |S_{2}^{\prime }\left(t\right)|=|a_{1}\pi cos\left(\pi t+b_{1}\right)+2a_{1}sin\left(b_{1}\right)+2\pi a_{2}cos\left(2\pi t+b_{2}\right)|\leq \\ |a_{1}\pi +2a_{1}+2\pi a_{2}\left|=\right|a_{1}\left(\pi +2\right)+2\pi a_{2}\left|\leq \right|a_{1}\left|\left(2+\pi \right)+2\pi \right|a_{2}| \end{array}$$

The inequality (Equation 24) leads to the conclusion that when $a_{1}$ and $a_{2}$ meet this inequality, the values of $b_{1}$ and $b_{2}$ can be anything. Values of *b* appear only as a phase shift of the trigonometric functions; therefore, they can be limited to the interval [0;2π] without missing any functions from the family. There is one additional degree of freedom when choosing values of *a*, because two values should meet only one inequality. Therefore, it is necessary to choose some arbitrary rule to limit that degree of freedom. Because the sine of amplitude $a_{2}$ has a period half that of the sine of amplitude $a_{1}$, it is assumed that it can have half the amplitude as well. This assumption leads to the following conjunction: (25)$$\begin{array}{c} a_{1(max)}\left(\pi +2\right)+2\pi a_{2(max)}=1\wedge a_{1(max)}=2a_{2(max)}\wedge \\ a_{1}\in \left(-a_{1(max)};a_{1(max)}\right)\wedge a_{2}\in \left(-a_{2(max)};a_{2(max)}\right) \end{array}$$

By solving equations, making part of that conjunction, it is possible to obtain the equation for $a_{2(max)}$ and $a_{1(max)}$: (26)$$\begin{array}{c} 1=a_{1(max)}\left(\pi +2\right)+2\pi a_{2(max)}=2a_{2(max)}\left(\pi +2\right)+2\pi a_{2(max)}=\\ a_{2(max)}\left(2\left(\pi +2\right)+2\pi \right)=a_{2(max)}\left(4\pi +4\right)=a_{2(max)}\left(4\left(\pi +1\right)\right) \end{array}$$(27)$$\begin{array}{c} a_{2(max)}=\frac{1}{\left(4(\pi +1)\right)}\Rightarrow a_{1(max)}=\frac{2}{4(\pi +1)}=\frac{1}{2(\pi +1)} \end{array}$$

Finally, we can write the required limits on the parameters of the function *f* to ensure that it meets the requirements (Equation 5) and (Equation 6): (28)$$\left\{\begin{array}{c} a_{1}\in \left[-\frac{1}{2(\pi +1)};\frac{1}{2(\pi +1)}\right]\\ b_{1}\in \left[0;2\pi \right]\\ a_{2}\in \left[-\frac{1}{4(\pi +1)};\frac{1}{4(\pi +1)}\right]\\ b_{2}\in \left[0;2\pi \right] \end{array}\right.$$

An example of curves obtained when using parameters that meet the inequality (Equation 28) is shown in Figure 3. And for comparison, an example of curves obtained using parameters that do not meet the inequality is shown in Figure 4. It is visible that when parameters violate the inequality, the function is not suitable to represent electrode shift due to the flexibility of the elastic band, because the resulting curve does not represent a function being a one-to-one mapping.

In this paper, we propose to modify the phantom’s geometry and take into account electrode disarrangement using the presented equations to modify electrode positions. An example of the phantom for healthy lungs and three different conditions with displaced electrodes is shown in Figure 2. The presented model allows for varying sizes and positions of its components to represent individual changes. In this work, the tissue’s permittivity and conductivity were specified for an excitation frequency of 100 MHz.

### 2.4. Training Datasets

When training a neural network for image reconstruction in CC-EIT, the main question is choosing the appropriate training dataset size. Due to the problem’s non-linearity and multidimensionality, the size of the required dataset should be relatively large. Previous research [10] has shown satisfactory results when using a dataset consisting of approximately 200,000 samples. Additionally, it is possible to increase neural network generalization ability by including samples with random ellipses in the training data. Using samples with random ellipses in the training data allowed for the reconstruction of unusual extreme cases, such as patients having only one lung, without explicitly including such cases into the training dataset.

In this work, we have generated 12 basic datasets that differ in size and level of electrode misposition. All datasets, representing human thorax images, contain four combinations, uniformly distributed, of health conditions mentioned above: healthy lungs, lungs affected by pleural effusion, lungs affected by pneumothorax, and lungs affected by hydropneumothorax, meaning both conditions are present at the same time. The presence of each condition is being drawn independently for each lung from a uniform distribution. Therefore, the probability of the lung being healthy is 25%, the probability of pneumothorax is 25%, the probability of pleural effusion is 25%, and the probability of pneumothorax and pleural effusion coexisting is 25%.

The electrode position shift due to the band elasticity is defined by a function (Equation 22) with parameters selected from the uniform distribution from ranges satisfying the condition (Equation 28). Additionally, the global and local position shifts are drawn using a truncated normal distribution with a mean value of zero. The absolute value of the shift is not bigger than the provided standard deviation. The obtained random value means the percentage of the angle between two adjacent electrodes. The levels of local and global mispositions are selected by specifying a standard deviation of a normal distribution. In our work, we have defined three levels of electrode disarrangement, which are shown in Table 1.

While generating a dataset, the positions and sizes of organs were randomly selected, similarly to the procedure described in [10]. If randomly selected positions caused organ overlaps, the procedure was repeated a maximum of 100 times. If a persistent overlap remained, the sample was dropped. That led to the dataset’s real size difference from the target number of samples. Examples of the thorax and random ellipsis images representing electrical conductivity distribution and corresponding capacitance measurements are shown, respectively, in Figure 5 and Figure 6. The descriptions of basic datasets are shown in Table 2.

Using 12 basic datasets, we have produced four datasets for actual neural network training and four datasets for network testing. The basic training datasets were concatenated, and the noise was added to the simulated capacitance measurements. The noise amplitude is determined to provide a signal-to-noise ratio (SNR) of 30 dB for opposite electrodes, which have the smallest capacitance values. In that case, the SNR for other electrodes will be higher, reflecting real data acquisition system behavior. Training the neural network using data with added noise in this manner showed very good performance in image reconstruction from real measurements performed by the EVT4 data acquisition system [42]. Because the training is conducted on a large dataset randomly split into training and validation parts, it is necessary to have an independent testing dataset [46]. In this work, we have generated small datasets with approximately 20000 samples containing thorax images to conduct testing of the trained network after completing the learning loop.

### 2.5. Neural Network Architecture and Training Procedure

In this work, we have used a U-Net-based [28] neural network, which was trained using a conditional generative adversarial network approach (cGAN) [47]. This method of training uses an additional neural network intended to distinguish true image samples and images as an output of the main network. This auxiliary network, called a discriminator, is a simple classifier that has an image at the input and outputs the probability of the image to be a true image. During the training, the discriminator learns better and better to detect images generated by the main network, called the generator. Meanwhile, the generator learns to deceive a discriminator and generate images that are indistinguishable from the real ones. This approach was first used to generate images of a certain class as a part of generative adversarial networks (GANs) [48]. However, later it was discovered that the same concept could be applied to image reconstruction, generating an image based on a given condition. In the case of electrical tomography, such a condition could be capacitance or impedance measurement vector [49,50].

A traditional cGAN approach requires a latent vector to be provided as a source of randomness to increase the variability of generated images. However, in the case of large images and a large size of the condition vector, it is more convenient to use dropout layers to make the network more flexible. Such an approach was proposed as part of the Pix2Pix technique [51] for the processing of high-resolution images. During our previous research [10] it showed its feasibility for the CC-EIT image reconstruction.

In this study, we continue to use the same architecture of the generator and discriminator networks [10]. In the generator, the measurement vector of size 992 is transformed into the image of size 64 × 64, which is then processed by six convolutional blocks and then by six deconvolutional blocks. At the end of the network, an additional convolutional layer is added to reduce the obtained 64-channel image into a one-channel image. Each convolutional block consists of a convolutional layer, an optional batch normalization layer, and a leaky ReLU activation. Each deconvolutional block consists of the deconvolutional layer, a batch normalization layer, an optional 50% dropout layer, and a leaky ReLU activation.

Discriminator network has two inputs: the image from the generator and the vector of measurements. At the output, there is the probability of the image being a correct reconstruction of the given measurements. The network consists of three convolutional blocks, the same as used in the generator, an additional convolutional layer, a batch normalization layer, a leaky ReLU activation, and an output convolutional block. Because the expected output is a probability, a sigmoid activation, which limits the value to be in the range [0;1], is applied to the output of the network: (29)$$\sigma \left(x\right)=\frac{1}{1+e^{-x}}$$

The classic approach for training a cGAN implies two-step training. At the first step, the generator is trained by locking the discriminator’s weights and setting its output to one. This procedure aims to make a generator to deceive a discriminator, generating images similar to the real ones. At the second step, the discriminator is trained. The generator’s weights are locked, and the generator’s output is used with a target probability of zero, while true images are used with a target probability of one. At this step, the discriminator is being taught to distinguish real images from those produced by a generator. The main challenge in this learning approach is to keep a balance between the generator and the discriminator. If the discriminator is too good and can always detect an image produced by the generator, then the generator has no possibility for improvement. At the same time, if the discriminator is too bad and always identifies the reconstructed image as a true image, then the generator also has no reason to generate better images.

It is possible to keep that balance by using different learning rates for the discriminator and the generator. Additionally, when the generator produces low-quality images, it is possible to modify the generator’s loss function by including terms influenced by real images. In this work, the loss function is modified in the following way: (30)$$L_{generator}=BCE\left(D\left(G\left(x\right)\right),1\right)+100\;\cdot \;L_{1}\left(G\left(x\right),y\right)+100\;\cdot \;L_{2}\left(G\left(x\right),y\right)$$
where *x*—capacitance measurements at generator’s input, *y*—real images corresponding to the given capacitance measurements, D—discriminator as a function, G—generator as a function, BCE, $L_{1}$ and $L_{2}$ are binary cross entropy, mean absolute error and mean squared error loss functions, respectively, defined by the following equations: (31)BCE(x,y)=y·log(x)+(1−y)·log(1−x)(32)$$L_{1}\left(x,y\right)=\left|x-y\right|$$(33)$$L_{2}\left(x,y\right)={\left(x-y\right)}^{2}$$

Managing learning rates allows us to fine-tune the training process and ensure that the discriminator has optimal classification ability. In this work, in addition to choosing the different learning rates for the discriminator and generator, we used cosine annealing warm restart learning rate scheduling. This scheduler was first proposed in [52] for the Stochastic Gradient Descent (SGD) optimizer. However, it also improves the network’s training using Adam optimizer [53]. In this work, we used the Adam optimizer, which requires the following parameters: starting learning rate, weight decay rate, and coefficients used for computing the running average of gradient and its square ($\beta_{1}$, $\beta_{2}$). Cosine annealing warm-up scheduler has two parameters: $T_{0}$ and $T_{mult}$, responsible for the shape of the cosine function used. The used learning rate scheduling parameters are shown in Table 3.

In all experiments, the number of epochs for training was chosen by using the early stopping approach. The maximum number of epochs was set to 150, but if there was no significant change in the loss function value during 30 epochs, then training was stopped.

### 2.6. Reconstruction Quality Assessment

To evaluate image reconstruction quality, it is possible to use pixel-dependent measures assessing the similarity of the reconstructed image and the ground truth image. In this paper, four measures were used: structural similarity index (SSIM), peak signal-to-noise ratio (PSNR), root-mean-square error (RMSE), and 2D correlation coefficient. SSIM is a metric comparing the similarity in the structure of the two images [54]. It has values in the range of [−1;1], where one means identical structure, zero means no similarity, and −1 means perfect anti-correlation. Using SSIM allowed us to understand how the structure of the reconstructed image is connected to the ground truth image. PSNR is a metric widely used to evaluate a noise level introduced by image compression, and it approximates human perception of reconstruction quality [55]. PSNR allows estimating the difference in the magnitude between ground truth and reconstructed images; higher values of metrics hint at a smaller difference between the images. RMSE allows for estimating the magnitude more accurately using a linear scale. The 2D correlation coefficient shows the interdependence between pixels of the ground truth image and the reconstructed image.

## 3. Results

To evaluate the influence of the variable electrode position on the image reconstruction quality using an artificial neural network, we have trained four neural networks using four different electrode disarrangement levels on the dataset containing simulated images of the human thorax and random ellipses. The number of training epochs varied due to employing an early stopping approach, and it is shown in Table 4 for each level alongside the training time in minutes. The training was conducted using an NVIDIA GeForce RTX 3090 GPU. The mean inference time was 1.8 ms on the same hardware, which allows the use of the proposed reconstruction method in a real-time application, providing a frame rate of up to 500 frames/s.

After training the networks, the reconstruction was conducted using testing datasets containing only human thorax images with four levels of electrode displacement. An example of image reconstruction of the sample with a large electrode disarrangement level conducted using the networks trained on datasets with zero and large electrode disarrangement is shown in Figure 7. Examples of the reconstruction for cases with medium and small electrode displacement levels are shown in Figure 8 and Figure 9, respectively.

It is clearly visible that when possible electrode displacement is not considered, the reconstruction quality is very poor. However, introducing the electrode displacement into the training dataset allows for achieving satisfactory reconstruction quality. The same effect is observable for all levels of electrode disarrangement considered.

Using the reconstructed images, we have computed the previously mentioned quality metrics’ mean values and standard deviation for each of 16 combinations of training and testing datasets. The resulting values of SSIM, PSNR, RMSE, and correlation coefficient are shown in Figure 10.

To ensure that results are consistent, we can consider the base case when no electrode displacement was introduced into the training and testing datasets. According to the comparison with our previous study [10] shown in Table 5, it is visible that we have managed to improve metric values for the base case. It is possible due to longer training, using an early stopping approach, and selecting the weights with the lowest loss function value.

Considering a medical application described in this paper, it is necessary to mention the importance of correct pathology identification using the reconstructed image. In our previous work [10], we had used the classifier network trained on ground truth images to detect expected health conditions from the reconstructed images. The performance of the classifier appeared to be very good, and in this paper, we assume that improving the technical image quality metrics should not decrease this classification ability.

## 4. Discussion

When considering electrical tomography as a medical imaging modality, it is necessary to address the issues caused by the nature of real-life measurements. Previous research found that electrode misplacement during measurements is a very important factor and strongly influences the quality of the obtained images. We have addressed three kinds of electrode displacement when using a flexible belt: rotation of the belt relative to the patient’s body, electrode shifting due to the flexibility of the belt, and local random electrode shifting along the belt circumference. Other possible kinds of displacement which could be investigated further are body shape differences between patients and body shape changes during measurement due to the patient’s breathing and involuntary movements.

In this study, we have investigated the influence of electrode displacement on the image reconstruction ability of the adversarial neural network in capacitively coupled EIT. Using a previously developed human thorax numerical model and a newly created algorithm for simulating electrode disarrangement due to electrode belt flexibility, we have trained four neural networks using different levels of electrode displacement. We have considered a base case with equidistant electrodes to ensure maintaining consistency with our previous study [10]. Due to a more careful neural network weight selection process, we have managed to improve the quality of the image reconstruction. That enabled us to continue investigating cases when samples with electrode displacement are present in the training and testing sets. We have found that when the neural network is trained on samples with only equidistant electrodes, the reconstruction quality of samples with displaced electrodes drops dramatically. However, introducing samples with small electrode displacement allowed for achieving very good reconstruction results even on samples with a large electrode disarrangement.

We have noticed that introducing samples with electrode displacement into the training dataset causes a slight decline in reconstruction quality, which is greater when a higher level of disarrangement is used. However, when a large level of disarrangement was used, the network was robust to any level of displacement. It is possible that, because the level of disarrangement was set as a standard deviation of the percent of the angle between adjacent equidistant electrodes and a normal distribution was used, the dataset with large disarrangement also included samples corresponding to lower levels of displacements. Therefore, the network could correctly learn how to reconstruct images when electrodes were shifted less than expected. Exposing the network to samples with a large variety of electrode misplacements led to a strong increase in robustness to electrode displacement.

Since this work was conducted solely using numerical simulation, it is not directly applicable to medical imaging, and further research using real measurements is required. It is necessary to verify the neural network’s image reconstruction ability from measurements conducted on real-life thorax phantoms, while the network could be trained using synthetic data. Development of such phantoms states a complicated task, because human tissue electrical conductivity is too high compared to available 3D printing materials and too low compared to existing conductive materials. Therefore, it is necessary to create a material with suitable electric properties and relatively strong mechanical properties by mixing different substances. There are different approaches to create such materials: mixing 3D printing-suitable plastic with graphite [56], mixing alumina powder with graphite and subsequent sintering [57], and using a mixture of gelatin and saline solution [58]. Another direction for further research is the investigation of other kinds of electrode displacement during data acquisition.

## 5. Conclusions

The CC-EIT usage for medical applications brings specific challenges connected with real-world variability. The previous studies have shown the feasibility of the CC-EIT approach for thoracic image reconstruction employing fixed geometry. In this study, we explored for the first time how electrode displacement affects machine-learning-driven image reconstruction. As a result we have found that electrode displacement in CC-EIT insignificantly worsens obtained image quality when using a dataset including cases with misplaced electrodes.

A dataset considering electrode displacement was prepared using the newly developed mathematical model, which is capable of producing displacements in a broader range than is physically possible. It was necessary in the case when the model assumes a constant measuring belt length due to the nature of image distortions resulting from the expected changes in the patient’s thorax size.

The promising results of the conducted work open the way to in-depth research on CC-EIT medical applications. Before starting tests in real clinical conditions, it is necessary to investigate neural network performance using the real phantoms. The feasibility of using a neural network trained on synthetic data for image reconstruction from the real data was previously shown on simple objects, and now the next step is to show this on phantoms reflecting human tissues.

## Figures and Tables

**Figure 1 sensors-25-06543-f001:**
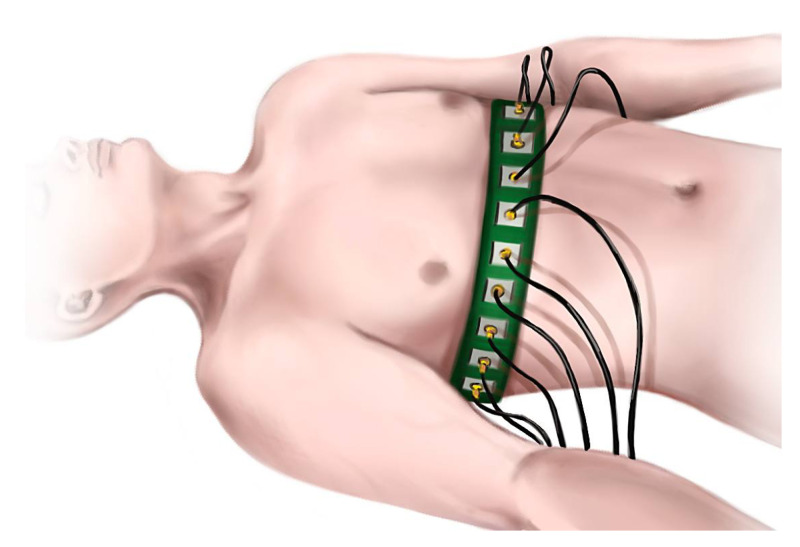
The artistic vision of the measuring belt on patient’s chest.

**Figure 2 sensors-25-06543-f002:**
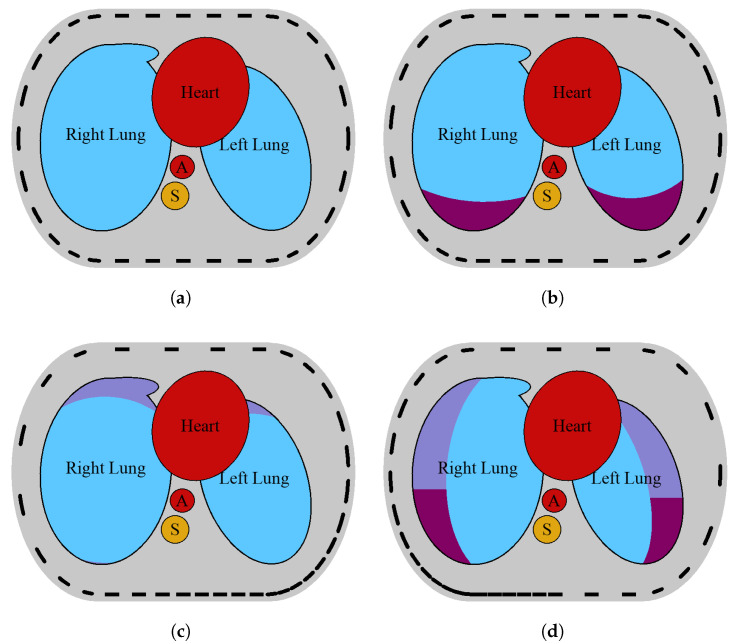
Model of a transversal slice of a human thorax with healthy lungs, heart, aorta (A), and spine (S) (**a**). Model of both lungs regionally affected by pneumothorax (**b**), pleural effusion with displaced electrodes (**c**), and hydropneumothorax (**d**). Pneumothorax and pleural effusion regions are shown, respectively, in violet and cherry [10]. The electrodes on the sensor strip may be evenly distributed (**a**) or variously displaced (**b**–**d**).

**Figure 3 sensors-25-06543-f003:**
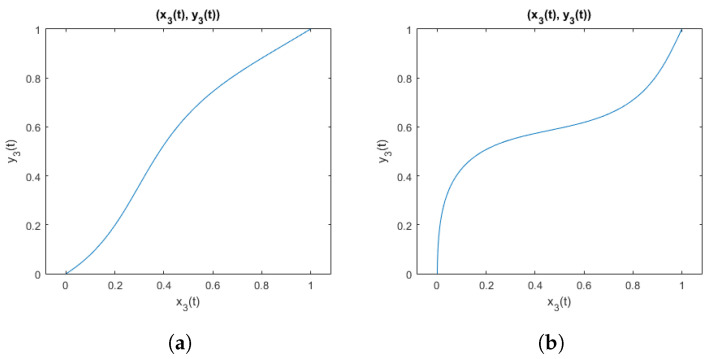
Curves from the family of functions defined by the inequality (Equation 28): (**a**) with parameters $\left(a_{1};b_{1};a_{2};b_{2}\right)=\left(-0.11;1.6;-0.033;1.5\right)$; (**b**) with parameters $\left(a_{1};b_{1};a_{2};b_{2}\right)=\left(0.1;0.2;0.1;0.1\right)$.

**Figure 4 sensors-25-06543-f004:**
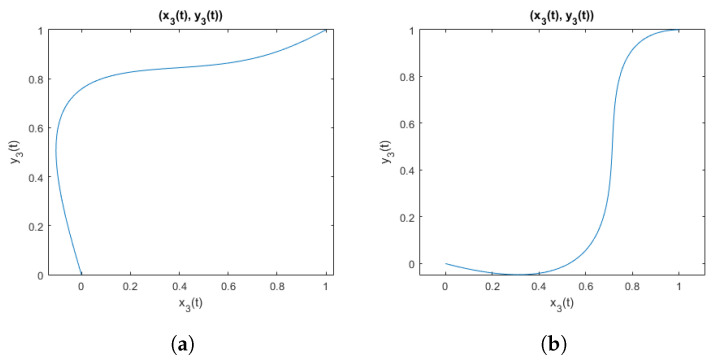
Curves from the family of functions not defined by the inequality (Equation 28): (**a**) with parameters $\left(a_{1};b_{1};a_{2};b_{2}\right)=\left(0.35;0.2;0.1;0.1\right)$; (**b**) with parameters $\left(a_{1};b_{1};a_{2};b_{2}\right)=\left(-0.7;1.4;0.04;1.4\right)$.

**Figure 5 sensors-25-06543-f005:**
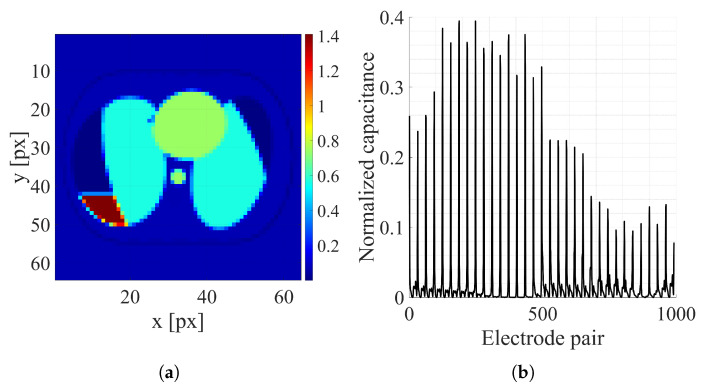
Thorax image sample: (**a**) Electrical conductivity distribution [S/m]. (**b**) Normalized capacitance measurements.

**Figure 6 sensors-25-06543-f006:**
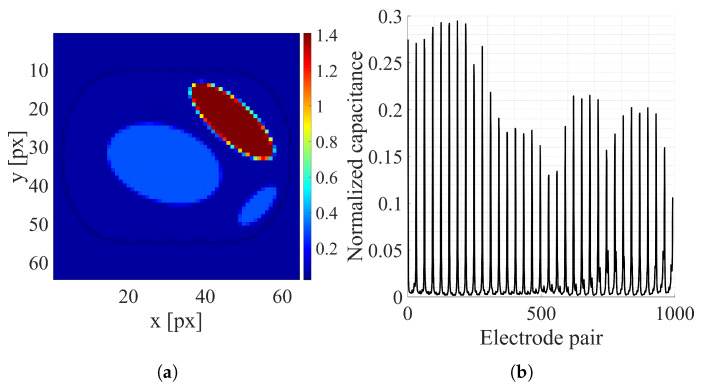
Random ellipsis sample: (**a**) Electrical conductivity distribution [S/m]. (**b**) Normalized capacitance measurements.

**Figure 7 sensors-25-06543-f007:**
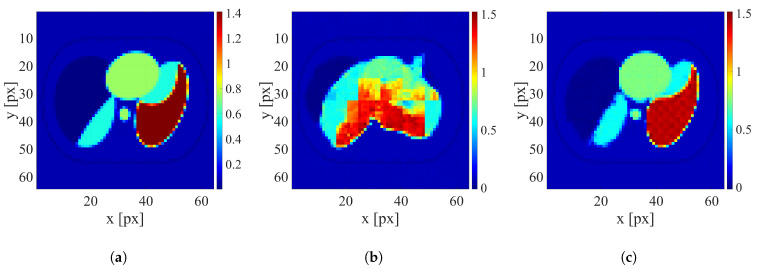
An example of image reconstruction with a large electrode disarrangement level. (**a**) Ground truth conductivity [S/m]. (**b**) Conductivity [S/m] reconstructed by the network trained on the dataset without electrode disarrangement. (**c**) Conductivity [S/m] reconstructed by the network trained on the dataset with a large electrode disarrangement.

**Figure 8 sensors-25-06543-f008:**
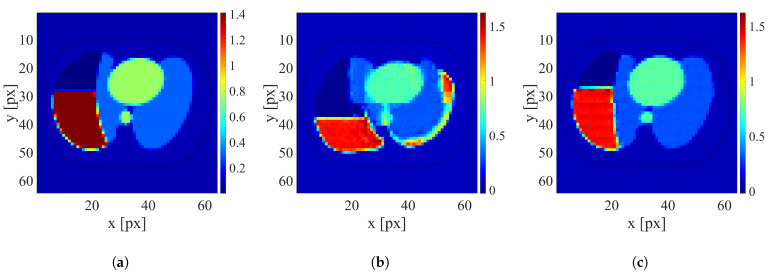
An example of image reconstruction with a medium electrode disarrangement level. (**a**) Ground truth conductivity [S/m]. (**b**) Conductivity [S/m] reconstructed by the network trained on the dataset without electrode disarrangement. (**c**) Conductivity [S/m] reconstructed by the network trained on the dataset with a medium electrode disarrangement.

**Figure 9 sensors-25-06543-f009:**
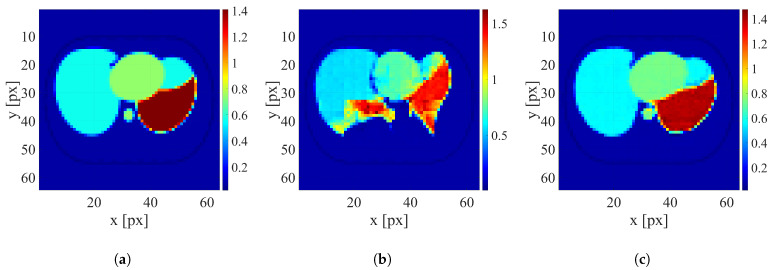
An example of image reconstruction with a small electrode disarrangement level. (**a**) Ground truth conductivity [S/m]. (**b**) Conductivity [S/m] reconstructed by the network trained on the dataset without electrode disarrangement. (**c**) Conductivity [S/m] reconstructed by the network trained on the dataset with a small electrode disarrangement.

**Figure 10 sensors-25-06543-f010:**
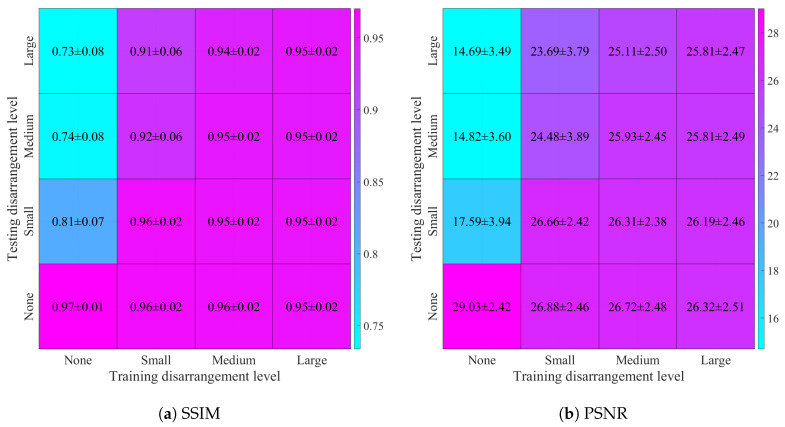
Image reconstruction quality metrics, calculated for four levels of electrode disarrangement used while training on the datasets containing simulated thorax images and random ellipses, and testing on the datasets containing simulated images of the human thorax with added noise (SNR = 30 dB): (**a**) structural similarity index distribution, (**b**) peak signal-to-noise ratio distribution, (**c**) root mean square error distribution, (**d**) 2D correlation coefficient distribution.

**Table 1 sensors-25-06543-t001:** Levels of electrode disarrangement.

Name	Local Shift Standard Deviation [%]	Global Shift Standard Deviation [%]
Small	2.5	7.5
Medium	5.0	12.5
Large	10.0	50.0

**Table 2 sensors-25-06543-t002:** Basic description of datasets.

Name	Content	Size	Electrode Disarrangement Level
Training datasets			
I	Thorax images	149,878	None
II	Random ellipses	48,857	None
III	Thorax images	149,912	Small
IV	Random ellipses	49,011	Small
V	Thorax images	149,912	Medium
VI	Random ellipses	49,011	Medium
VII	Thorax images	149,912	Large
VIII	Random ellipses	49,011	Large
Testing datasets			
IX	Thorax images	19,985	None
X	Thorax images	19,988	Small
XI	Thorax images	19,988	Medium
XII	Thorax images	19,988	Large

**Table 3 sensors-25-06543-t003:** Learning rate scheduling parameters used while training the neural network.

Parameter	Generator	Discriminator
Starting learning rate	$10^{-3}$	$10^{-6}$
Weights decay	$10^{-1}$	$10^{-1}$
$\beta_{1}$	0.5	0.5
$\beta_{2}$	0.999	0.999
$T_{0}$	$10^{3}$	$10^{3}$
$T_{mult}$	2	2

**Table 4 sensors-25-06543-t004:** Number of training epochs using the dataset containing simulated images of the human thorax with added low-level noise (SNR = 30 dB).

Electrode disarrangement level	None	Small	Medium	Large
Number of epochs	93	150	141	148
Training time [min]	63	104	96	99

**Table 5 sensors-25-06543-t005:** Comparison of image quality reconstruction metrics for the base case when the network was trained and tested without electrode disarrangement with the previous study.

Metric	Previously Achieved in [10]	This Study Base Case
RMSE	9.58±4.43	8.06±2.82
PSNR	26.8±2.98	29.03±2.42
SSIM	0.87±0.03	0.97±0.01
2D Correlation coefficient	0.98±0.03	0.99±0.01

## Data Availability

Dataset available on request from the authors.

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
