# Peer review of "Machine Learning-Based Image Reconstruction in Wearable CC-EIT of the Thorax: Robustness to Electrode Displacement"

_sensors, 2025, doi:10.3390/s25216543_

Round 1

Reviewer 1 Report

Comments and Suggestions for Authors

The paper proposed a parametric function with trigonometric terms to map electrode positions under elastic band deformation. Designed a cGAN-based reconstruction network with hybrid loss and thus enhanced robustness to electrode displacement. Before the paper is accepted for publication, the following concerns should be addressed:

  1. The paper only adds 30dB Gaussian noise to capacitance measurements, ignoring complex noise in real CC-EIT systems (e.g. power-line interference, electrode contact noise). A single noise model cannot verify performance in real electromagnetic environments.
  2. The current cGAN generator is data-driven and does not explicitly utilize CC-EIT electromagnetic priors. This may not only lead to physically implausible results but also increases the computational burden.
  3. All experiments rely on simulated data from numerical phantoms, without validation using real hardware-acquired capacitance data. Some of the real world problems may absent in simulated data.
  4. The current model targets static electrode displacement, but in practice, respiratory-induced thoracic deformation causes periodic electrode movement. Static models fail to handle such dynamic signals, leading to inter-frame inconsistency in reconstructed images.
  5. The paper tests single pathologies but not multi-pathology coexistence, which is common in clinical practice. The coupling effect of conductivity differences across multiple pathological regions and electrode displacement may exacerbate reconstruction errors.

Reviewer 2 Report

Comments and Suggestions for Authors

The research paper entitled "Machine Learning-Based Image Reconstruction in Wearable
CC-EIT of the Thorax: Robustness to Electrode Displacement", mainly discusses the process of the reconstruction of EIT images using a deep learning GAN-based approach. This is a well-written and highly relevant study addressing a critical challenge in wearable CC-EIT: electrode displacement. 

The use of a cGAN-based approach is appropriate, and the synthetic data generation is rigorous. The results clearly demonstrate that including displacement in training significantly improves reconstruction robustness.

Major Points:

While the simulation methodology is strong, the practical impact of this work would be greatly enhanced by validation on real-world data, such as from a physical thorax phantom. This would help confirm the model's performance under realistic conditions. This should be, at least, clearly discussed in the future work section, although doing it in this research would significantly strengthen the article.

The paper focuses heavily on technical image quality metrics (SSIM, PSNR, etc.) but doesn't address how these improvements translate to diagnostic accuracy or clinical outcomes. This gap between technical performance and medical utility is worth noting.

Moreover, a comparison with alternative approaches to electrode displacement would be valuable to the article.

Also, no computational cost analysis for real-time feasibility.

Would your approach generalize across different body types or pathological conditions? Add a paragraph to discuss this to clarify this point to the readers.

Finally, while the authors develop a physics-based framework, the specific trigonometric approach (equations 5-28) is quite complex. The paper would benefit from justifying why this particular mathematical formulation is necessary compared to simpler elastic deformation models.

Minor Points:

P2, Line ~72: “in-verse” → “inverse”

P5, Line ~200: “There-fore” → “Therefore”

Make sure to correct these and similar typographical issues in the manuscript.

I highly suggest a resubmission for this interesting article.

Round 2

Reviewer 1 Report

Comments and Suggestions for Authors

The authors have addressed all  my concerns, the paper is ready for publication

Reviewer 2 Report

Comments and Suggestions for Authors

I have no further comments